# USING ONTOLOGIES TO IMPROVE PERFORMANCE IN MASSIVELY MULTI-LABEL PREDICTION

## ABSTRACT

Massively multi-label prediction/classification problems arise in environments like health-care or biology where it is useful to make very precise predictions. One challenge with massively multi-label problems is that there is often a long-tailed frequency distribution for the labels, resulting in few positive examples for the rare labels. We propose a solution to this problem by modifying the output layer of a neural network to create a Bayesian network of sigmoids which takes advantage of ontology relationships between the labels to help share information between the rare and the more common labels. We apply this method to the two massively multi-label tasks of disease prediction (ICD-9 codes) and protein function prediction (Gene Ontology terms) and obtain significant improvements in per-label AUROC and average precision.

## 1 INTRODUCTION

In this paper, we study general techniques for improving predictive performance in massively multi-label classification/prediction problems in which there is an ontology providing relationships between the labels. Such problems have practical applications in biology, precision health, and computer vision where there is a need for very precise categorization. For example, in health care we have an increasing number of treatments that are only useful for small subsets of the patient population. This forces us to create large and precise labeling schemes when we want to find patients for these personalized treatments.

One large issue with massively multi-label prediction is that there is often a long-tailed frequency distribution for the labels with a large fraction of the labels having very few positive examples in the training data. The corresponding low amount of training data for rare labels makes it difficult to train individual classifiers. Current multi-task learning approaches enable us to somewhat circumvent this bottleneck through sharing information between the rare and cofmmon labels in a manner that enables us to train classifiers even for the data poor rare labels (Caruana, 1997).

In this paper, we introduce a new method for massively multi-label prediction, a Bayesian network of sigmoids, that helps achieve better performance on rare classes by using ontological information to better share information between the rare and common labels. This method is based on similar ideas found in Bayesian networks and hierarchical softmax (Morin & Bengio, 2005). The main distinction between this paper and prior work is that we focus on improving multi-label prediction performance with more complicated directed acyclic graph (DAG) structures between the labels while previous hierarchical softmax work focuses on improving runtime performance on multi-class problems (where labels are mutually exclusive) with simpler tree structures between the labels.

In order to demonstrate the empirical predictive performance of our method, we test it on two very different massively multi-label tasks. The first is a disease prediction task where we predict ICD-9 (diagnoses) codes from medical record data using the ICD-9 hierarchy to tie the labels together. The second task is a protein function prediction task where we predict Gene Ontology terms (Ashburner et al., 2000; Carbon et al., 2017) from sequence information using the Gene Ontology DAG to combine the labels. Our experiments indicate that our new method obtains better average predictive performance on rare labels while maintaining similar performance on common labels.

## 2 METHODS

### 2.1 PROBLEM SETUP

The goal of multi-label prediction is to learn the distribution $P(L|X)$ which gives the probability of an instance $X$ having a label $L$ from a dictionary of $N$ labels. We are particularly interested in the case where there is an ontology providing superclass relationships between the labels. This ontology consists of a DAG where every label $L$ is a node and every directed edge from $L_i$ to $L_j$ indicates that the label $L_i$ is a superclass of the label $L_j$. Figure 1 gives corresponding example simplified subgraphs from both the ICD-9 hierarchy and the Gene Ontology DAG. We define $parents(L)$ to be the direct parents of $L$. We define $ancestors(L)$ to be all of the nodes that have a directed path to $L$.

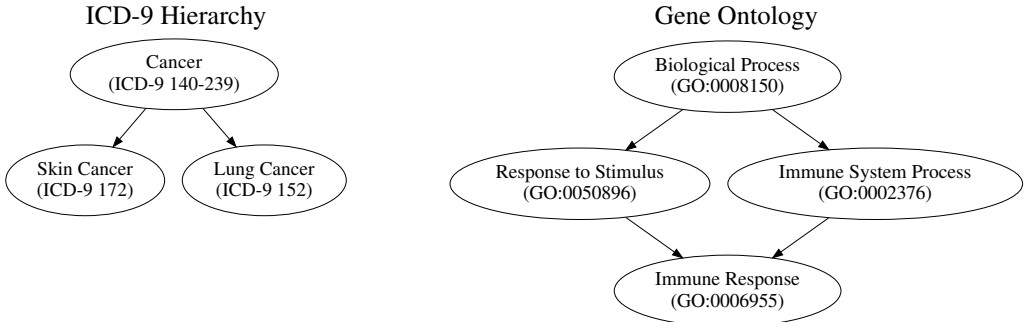

Figure 1: Example simplified graphs showing superclass relationships from the ICD-9 hierarchy and the Gene Ontology DAG.

The classical approach for solving this problem is to learn separate functions for each label. This transforms the problem into $N$ binary prediction problems which can each be solved with standard techniques. The main issue with this approach is that it is less sample efficient in that it does not share information between the labels. A more sophisticated approach is to use multi-task learning techniques to share information between the individual label-specific binary classifiers. One approach for doing this with neural networks is to introduce shared layers between the different binary classifiers. The resulting output layer is a flat structure of sigmoid outputs, with each sigmoid output representing one $P(L|X)$. This reduces the number of parameters needed for every label and allows information to be shared among the labels (Caruana, 1997). However, even with this weight sharing, the final output layer still needs to be learned independently for each label.

### 2.2 BAYESIAN NETWORK FACTORIZATION

We propose a modification of the output layer by constructing a Bayesian network of sigmoids in order to use the ontology to share additional information between labels in a more guided way. The general idea is that we assume that the probability of our labels follows a Bayesian network (Pearl, 1988) with each edge in the ontology representing an edge within the Bayesian network. This, along with the fact that the edges denote superclasses, enables us to factor the probability of a label into several conditional probabilities.

$$P(L|X) = P(L, ancestors(L)|X)$$

As the edges denote superclasses, having a child label implies having every ancestor

$$= \prod_{\ell \in \{L\} \cup ancestors(L)} P(\ell|X, parents(\ell))$$

From Baysian network assumption on the subgraph consisting of $L$ and $ancestors(L)$ (Pearl, 1988)

We are now able to learn the conditional probability distributions $P(L|X, parents(L))$ for every label in the ontology and use the above formula to reconstruct the final target probabilities

$P(L|X)$. Consider the example simplified ICD-9 graph in Figure 1. For this graph, we would learn $P(Cancer|X)$, $P(LungCancer|Cancer, X)$, and $P(SkinCancer|Cancer, X)$. We would then be able to compute $P(LungCancer|X) = P(Cancer|X) \times P(LungCancer|Cancer, X)$.

The intuition of why this factoring might be useful is that it enables the transferring of knowledge from more common higher-level labels to more rare lower-level labels. Consider the case where $L$ is very rare. In that case it is difficult to learn $P(L|X)$ directly due to the small amount of training data. However, the decomposed version $\prod_{\ell \in \{L\} \cup ancestors(L)} P(\ell|X, parents(\ell))$ includes classifiers from the ancestors of $L$ that have more training data and might be easier to learn. This factoring allows additional signal from the better trained higher-level labels to feed directly into the probability computation for the rare leaf $L$. If we can rule out one of the higher-level labels, we can also rule out a lower-level label. For example, consider the ICD-9 graph illustrated in Figure 1. We might not have enough patients with lung cancer to directly learn an optimal $P(LungCancer|X)$. However, we can pool all of our cancer patients to learn a hopefully more optimal $P(Cancer|X)$. We can then use our Bayesian network factoring to incorporate the better trained $P(Cancer|X)$ classifier in our calculation for $P(LungCancer|X)$. In our experiments we show that this intuition plays out in practice through improved performance on rare labels.

The Bayesian network assumption plays an important role in allowing us to factor the probabilities in this manner. In order to perform our factoring, we must assume that every subgraph of the ontology consisting of the nodes $\{L\} \cup ancestors(L)$ correctly represents a Bayesian network for the label probability distribution. These subgraphs are only correct Bayesian networks if the probability of every label $L$ is conditionally independent of the probabilities of non-descendent labels given the parent labels and $X$ (Russell & Norvig, 2009). This might seem somewhat limiting, but there are two reasons why this assumption is weaker than it might appear. First, we only require a Bayesian network to be correct for the subgraphs of the form $\{L\} \cup ancestors(L)$. This is true because we only consider the nodes $\{L\} \cup ancestors(L)$ when we do our factoring. This is a significantly weaker assumption than requiring the entire graph to follow a Bayesian network. One direct application of this is that every tree ontology can meet this assumption. The proof for this is that every $\{L\} \cup ancestors(L)$ subgraph of a tree is a simple chain. A simple chain is not able to violate the conditional independence assumption behind Bayesian networks because it has no non-descendent nodes that are not already ancestors. Ancestor nodes are always conditionally independent with the label given the parents because the edges represent superclasses and thus either the ancestors are always present if the parent i present or the label is always not present if the parent is not present. The second reason why this assumption is weaker than it might appear is that we only require conditional independence given a particular instance $X$. As an illustrative example, consider the two ICD-9 labels of male breast cancer (ICD-9 175) and female breast cancer (ICD-9 174). Male breast cancer and female breast cancer are trivially not conditionally independent due to the gender qualifier making them mutually exclusive. However, male breast cancer and female breast cancer become conditionally independent once you condition on the gender of the patient. Thus conditioning on the exact instance $X$ enables more conditional independence than would otherwise be available. Nevertheless, even with these caveats, there will be some circumstances in which this conditional independence assumption is violated. In these situations, our factoring is not valid and our computed product $\prod_{\ell \in \{L\} \cup ancestors(L)} P(\ell|X, parents(\ell))$ might diverge from the actual $P(L|X)$. Yet, even in these situations, the resulting scores can still be empirically useful. We demonstrate that this is the case in our experiments by showing performance improvements in a protein function prediction task that almost assuredly violates this conditional independence assumption.

## 2.3 Modeling The Probabilities With Sigmoid

There are many potential ways in which the conditional probabilities $P(L|X, parents(L))$ could be modeled. We exclusively focus on modeling these probabilities using a sigmoid function computed on logits from neural networks. We define an encoder neural network for every task that takes in the input $X$ and returns a fixed-length representation of the input. We also define a fixed-length embedding for every label $L$ by constructing an output embedding matrix such that $e_L$ is the embedding for $L$. This encoder and label embedding then allow us to model $P(L|X, parents(L))$ as $\sigma(encoder(X) \cdot e_L)$, where $\sigma$ indicates the sigmoid function and $\cdot$ indicates a dot product. Note that $parents(L)$ is not used in this formula. This is because there is a unique set of parents for every label $L$, so there is no need to have distinct $e_L$ vectors for different sets of parents. We can then train

$P(L|X, parents(L))$ by using cross entropy loss on patients who have all the labels in $parents(L)$. Note that we explicitly do not train each of the conditional probabilities on every patient. We can only train the conditional probabilities on patients who satisfy the conditional requirement of having the parent labels. This does not change the number of positive examples for each classifier, but it does significantly reduce the number of negative examples for the lower level classifiers.

For example, consider the ICD-9 subgraph shown in Figure 1. In this situation, we have three labels and thus need to learn three conditional probabilities: $P(Cancer|X)$, $P(LungCancer|Cancer, X)$ and $P(BreastCancer|Cancer, X)$. We have three labels, so our label embedding matrix consists of $e_{Cancer}$, $e_{LungCancer}$ and $e_{BreastCancer}$. We can now compute $P(LungCancer|X)$ and $P(BreastCancer|X)$ as follows:

$$P(LungCancer|X) = P(LungCancer|Cancer, X) \times P(Cancer|X)$$
$$= \sigma(encoder(X) \cdot e_{LungCancer}) \times \sigma(encoder(x) \cdot e_{Cancer})$$

$$P(BreastCancer|X) = P(BreastCancer|Cancer, X) \times P(Cancer|X)$$
$$= \sigma(encoder(X) \cdot e_{BreastCancer}) \times \sigma(encoder(X) \cdot e_{Cancer})$$

As a baseline, we also train models with a normal flat sigmoid output layer. In these models we directly learn $P(L|X)$ for each label. Similar to the conditional probabilities, we can define these probabilities as a sigmoid of the output from a neural network. We define $P(L|X)$ to be $\sigma(encoder(X) \cdot e_L)$. We can then train $P(L|X)$ using cross entropy loss on all patients.

## 3 EXPERIMENTAL SETUP

We evaluated the predictive performance of our method on two very different massively multi-label problems. We consider the task of predicting future diseases for patients given medical history in the form of ICD-9 codes and the task of predicting protein function from sequence data in the form of Gene Ontology terms. In this section, we introduce the datasets, encoders and baselines used for each problem.

### 3.1 DISEASE PREDICTION

#### 3.1.1 PROBLEM

One of our experiments consists of predicting diseases in the form of ICD-9 codes from electronic medical record (EMR) data. We have two years and nine months of data covering 2013, 2014, and the first nine months of 2015. We use two years of history to predict which ICD-9 codes will appear in the following nine months. The problem setup for this experiment closely matches the setup in Miotto et al. (2016). We use a large insurance claims dataset from [redacted to preserve anonymity] for modeling. Our claims data consists of diagnoses (ICD-9), medications (NDC), procedures (CPT), and some metadata such as age, gender, location, general occupation, and employment status. We restrict our analysis to patients who were enrolled during 2013, 2014 and January 2015.

We have 15.7 million patients, of which a random 5% are used for validation and 5% are used for testing. This dataset is quite large, much larger than what is usually available in a hospital. Thus we consider two cases of this problem. The "high data case" is where we use all remaining 14.1 million patients for training. The " low data case" consists of training with a 2% random sample of 281,874 patients and is much closer in size to normal hospital datasets (Choi et al., 2017; Avati et al., 2017).

Our target label dictionary for this task consists of all leaf ICD-9 billing codes that appear at least 5 times in the training data. We only predict leaf codes as those are the only codes allowed for billing and thus the only ICD-9 codes that records are annotated with. This results in a dictionary of 6,902 codes for the small disease prediction task and 12,533 codes for the large disease prediction task. We use the ICD-9 hierarchy included in the 2018AA UMLS release (Bodenreider, 2004) in order to construct relationships between the labels for our method. We additionally use the CPT and ATC ontologies included in the 2018AA for our encoder.

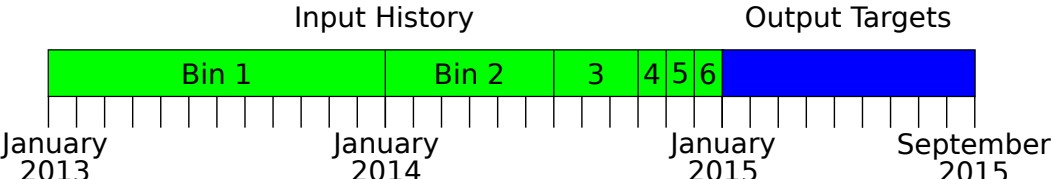

Figure 2: The partitioning of the patient timelines into input history and output prediction labels as well as the subpartioning of the input history into time-bins. Each tick on the x-axis represents one month. The first two years of information is used as input and the final nine months is used to generate output prediction labels. These first two years are subdivided into six bins of the following lengths for featurization: one year, six months, three months, one month, one month, and one month.

### 3.1.2 ENCODER DESCRIPTION

For our encoder, we use a feed-forward architecture inspired by Avati et al. (2017). As in their model, we split our two years of data into time-sliced bins. For each time slice, we find all the ICD-9, NDC and CPT codes that the patient experienced during the time slice. Figure 2 details the exact layout of each time bin. We also add a feature for every higher-level code in the ICD-9, ATC and CPT ontologies that indicates whether the patient had any of the descendants of that particular code within the time slice. This expanded rollup scheme is structurally very similar to the subword method introduced in Bojanowski et al. (2017). The weights for these input embeddings are tied to the output embedding matrix used in our output layers. We summarize the set of embeddings for each time bin using mean pooling. We also construct mean embedding for the metadata by feeding the metadata entries through an embedding matrix followed by mean pooling. Finally, we concatenate the means from each timeslice with the mean embeddings from the metadata and feed the resulting vector into a feedforward neural network to compute a final patient embedding.

These neural network models are trained with the Adam optimizer. The hyperparameters such as the learning rate, layer size, non-linearity, and number of layers are optimized using a grid search on the validation set. Appendix A.1 has details on the space searched as well as the best hyper-parameters for both the normal sigmoid and Bayesian network sigmoid models.

Finally, as a further baseline, we also train logistic regression models individually for several rare ICD-9 codes. These models are trained on a binary matrix where each row represents a patient and each column represents an ICD-9 code, NDC code, CPT code, or metadata element. A particular row and column element is set to 1 whenever a patient has that particular item in the metadata or during the two years of provided medical history. These logistic regression models are regularized with L2 with a lambda optimized using cross-validation. One particular issue with training individual models on rare codes is that the dataset is distinctly unbalanced with vastly more negative examples than positive examples. We deal with this issue by subsampling negative examples so that the ratio of positive and negative samples is 1:10.

### 3.2 PROTEIN FUNCTION PREDICTION

### 3.2.1 PROBLEM

For our other experiment, we predict protein functions in the form of Gene Ontology (GO) terms from sequence data. We focus only on human proteins that have at least one Gene Ontology annotation. Our features consist of amino acid sequences downloaded from Uniprot on July 27, 2018 (Consortium, 2017). For our labels, we use the human GO labels which were generated on June 18, 2018. After joining the labels with the sequence data, we have a total of 15,497 annotated human protein sequences. A random 80% of the sequences are used for training, 10% are using for validation, and a final 10% are used for final testing.

In this task we predict all leaf and higher level GO terms that appear at least 5 times in the training data. This results in a target dictionary of 7,751 terms. We construct relationships between these labels using the July 24, 2018 release of the GO basic ontology.

### 3.2.2 Encoder Description

We use Kim (2014)'s 1-D CNN based encoder to encode our protein sequence information. We treat every letter in the alphabet as a word and encode each of those letters with an embedding size of size 26. We then apply a 1-D convolution with a window size of 8 over the embedded sequence. A fixed-length representation of the protein is then obtained by doing max-over-time pooling. This representation is finally fed through a ReLU and one fully connected layer. The resulting fixed dimension vector is the encoded protein. For regularization, we add dropout before the convolution and fully connected layer.

Following previous work, we also consider generating features using sequence alignment (Kulmanov et al., 2018). We use version 2.7.1 of the BLAST tool to find the most similar training set protein for every protein in our dataset (Wheeler et al., 2007). We then use this most similar protein to augment our protein encoder by adding a binary feature which signifies if the most similar protein has the particular term we are predicting.

These CNN models are trained with Adam. Hyperparameters such as learning rate, number of filters, dropout, and the size of the final layer are optimized using a grid search on the validation set. See Appendix A.1 for a full listing of the space searched as well as the best hyperparameters for both the flat sigmoid and Bayesian network of sigmoids models.

As a further baseline, we also consider using the BLAST features alone for predicting protein function. This model simply consists of a 1 if the most similar protein has the target term or a 0 otherwise.

For these protein models, we also consider one final baseline where we take our flat sigmoid model and weight labels according to the inverse square root of their frequency. This weighting scheme is based off the subsampling scheme from Mikolov et al. (2013). Unfortunately, this baseline did not seem to perform well on rare words so we did not consider it for the disease case and our more general analysis. The results for this baseline can be found in Appendix A.3.

## 4 Results

Figure 3 shows frequency binned per-label area under the receiver operating characteristic (AUROC) and average precision (AP) for less frequent labels that have at most 1,000 positive examples. See Appendix A.2 for the exact numerical results which include 95 % bootstrap confidence intervals generated through 500 bootstrap samples of the test set. As shown in Figure 4, these less frequent labels cover a majority of the labels within each dataset. Our results indicate that the Bayesian network of sigmoid output layer has better AUROC and average precision for rare labels in all three tasks, with the effect diminishing with increasing numbers of positive labels. This effect is especially strong in the average precision space. For example, the Bayesian network of sigmoid models obtain 187%, 28.5% and 17.9% improvements in average precision for the rarest code bin (5-10 positive examples) over the baseline models for the small disease, large disease and protein function tasks, respectively. This improvement persists for the next rarest bin (11-25 positive examples), but decreases to 89.2%, 10.7% and 11.1%. This matches our previous intuition as there is no need to transfer information from more general labels if there is enough data to model $P(L|X)$ directly.

Table 1 compares micro-AUROC and micro-AP on all labels for all three tasks. The benefits of the Bayesian sigmoid output layer seem much more limited and task specific in this setting. We do not expect significantly better results in the micro averaged performance case because the micro results are more dominated by more frequent codes and the Bayesian network of sigmoids is only expected to help when $P(L|X)$ does not have enough data to be modeled directly. The Bayesian network of sigmoids output layer provides better AUROC and AP for the disease prediction task, but suffers from worse performance in the protein function task. One possible explanation for this discrepancy is that our Bayesian network assumption is guaranteed to be correct in the disease prediction task due to the tree structure of the ontology, but might not be correct in the protein function task with its more complicated DAG ontological structue. It is possible that minor violations of the Bayesian network assumption in the protein function prediction task cause the overall performance to be worse on the more common code compared to the flat sigmoid decoder.

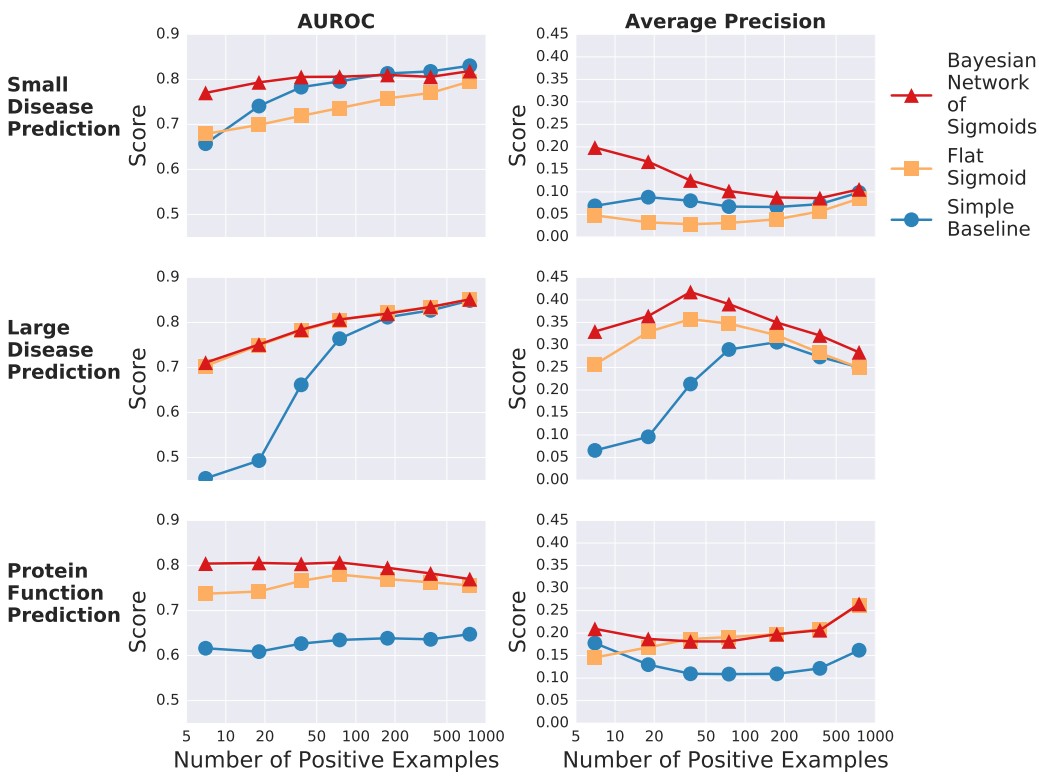

Figure 3: Frequency binned per-label AUROC and average precision (AP) for less frequent labels with at most 1,000 positive examples. AUROC and AP are calculated independently for each individual label. Labels are then grouped into bins determined by the number of positive samples per label and average statistics are computed for each bin. The x-axis is in log-scale and represents the number of possible examples for the center of each bin. Each line represents the type of model, with the baseline model differing between the disease prediction and protein function prediction tasks.

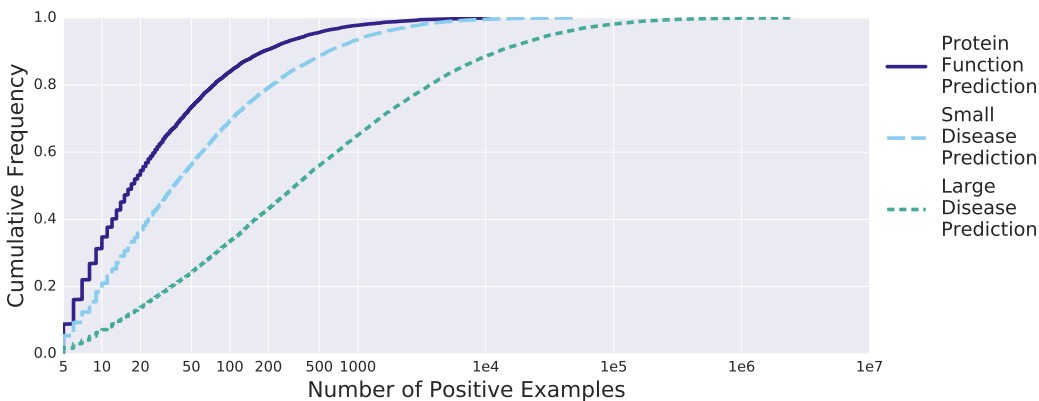

Figure 4: The cumulative frequency distribution for the target labels in the various tasks. The x-axis is in log scale.

## 5 RELATED WORK

There is related work on improved softmax variants, predicting ICD-9 codes, predicting Gene Ontology terms and combining ontologies with Bayesian networks.

Table 1: Micro-AUROC and micro-average precision (AP) results on the various tasks.

| Model | Small Disease | | Large Disease | | Protein Function | |
|---|---|---|---|---|---|---|
| | AUROC | AP | AUROC | AP | AUROC | AP |
| Flat Sigmoid | 0.951 | 0.209 | **0.982** | 0.262 | **0.945** | **0.436** |
| Bayesian Network of Sigmoids | **0.960** | **0.220** | **0.982** | **0.269** | 0.935 | 0.430 |

**Improved softmax variants.** There has been a wide variety of work focusing on trying to come up with improved softmax variants for use in massively multi-class problems such as language modeling. This prior work primarily differs from this work in that it focuses exclusively on the multi-class case with a tree structure connecting the labels. Multi-class is distinct from multi-label in that multi-class requires each item to only have one label while multi-label allows multiple labels per item. Most of this work focuses around trying to improve the training time for the expensive softmax operation found in multi-class problems such as large-vocabulary language modeling. The most related of these variants fall under the hierarchical softmax family. Hierarchical softmax from Morin & Bengio (2005) (and related versions such as class based softmax from Goodman (2001) and adaptive softmax from Grave et al. (2016)) focuses on speeding up softmax by using a tree structure to decompose the probability distribution.

**Disease prediction.** Previous work has also explored the task of disease prediction through predicting ICD-9 codes from medical record data (Miotto et al., 2016; Choi et al., 2017; 2015). GRAM from Choi et al. (2017) is a particularly relevant instance which uses the CCS hierarchy to improve the encoder, resulting in better predictions for rare codes. Our work differs from GRAM in that we improve the output layer while GRAM improves the encoder.

**Protein function prediction.** Protein function prediction in the form of Gene Ontology term prediction has been considered by previous work (Kulmanov et al., 2018; Lan et al., 2013; Cao et al., 2017). DeepGO from Kulmanov et al. (2018) is the most similar to the approach taken by this paper in that it uses a CNN on the sequence data to predict Gene Ontology terms. It also uses the ontology in that it creates a multi-task neural network in the shape of the ontology. Our work differs from DeepGO in that we focus on the rarer terms and we only modify the output layer.

**Combining ontologies with Bayesian networks.** Phrank from Jagadeesh et al. (2018) is an algorithm for computing similarity scores between sets of phenotypes for use in diagnosing genetic disorders. Like this paper, Phrank constructs a Bayesian network based on an ontology. This work differs from Phrank in that we focus on the supervised prediction task of modeling the probability of a label given an instance while Phrank focuses on the simpler task of modeling the unconditional probability of a label (or set of labels).

## 6 CONCLUSION

This paper introduces a new method for improving the performance of rare labels in massively multi-label problems with ontologically structured labels. Our new method uses the ontological relationships to construct a Bayesian network of sigmoid outputs which enables us to express the probability of rare labels as a product of conditional probabilities of more common higher-level labels. This enables us to share information between the labels and achieve empirically better performance in both AUROC and average precision for rare labels than flat sigmoid baselines in three separate experiments covering the two very different domains of protein function prediction and disease prediction. This improvement in performance for rare labels enables us to make more precise predictions for smaller label categories and should be applicable to a variety of tasks that contain an ontology that defines relationships between labels.

ACKNOWLEDGMENTS

This section has been redacted to preserve anonymity.

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

# A APPENDIX

## A.1 HYPERPARAMETER GRID AND BEST HYPERPARAMETERS

Table 2: Hyperparameter space explored for small disease prediction

| Hyperparameter Name | Values Explored |
|---|---|
| Learning Rate | $[10^{-2}, 10^{-3}, 10^{-4}, 10^{-5}, 10^{-6}]$ |
| Embedding Size | [64, 128, 256, 512] |
| Number Of Additional Layers | [0, 1, 2] |
| Additional Layer Size | [128, 256, 512] |
| Activation function | [identity, ReLU] |
| Shared Weights | [False, True] |

Table 3: Best hyperparameters for small disease prediction

| Hyperparameter Name | Flat Sigmoid | Bayesian Network of Sigmoids |
|---|---|---|
| Learning Rate | $10^{-5}$ | $10^{-4}$ |
| Embedding Size | 512 | 256 |
| Number Of Additional Layers | 0 | 0 |
| Layer Size | N/A | N/A |
| Activation function | identity | identity |
| Shared Weights | True | True |

Table 4: Hyperparameter space explored for large disease prediction

| Hyperparameter Name | Values Explored |
|---|---|
| Learning Rate | $[10^{-3}, 10^{-4}, 10^{-5}]$ |
| Embedding Size | [256, 512] |
| Number Of Additional Layers | [0, 1, 2] |
| Additional Layer Size | [128, 256, 512] |
| Activation function | [identity, ReLU] |
| Shared Weights | [False, True] |

Table 5: Best hyperparameters for large disease prediction

| Hyperparameter Name | Flat Sigmoid | Bayesian Network of Sigmoids |
|---|---|---|
| Learning Rate | $10^{-4}$ | $10^{-4}$ |
| Embedding Size | 512 | 512 |
| Number Of Additional Layers | 0 | 0 |
| Layer Size | N/A | N/A |
| Activation function | ReLU | identity |
| Shared Weights | True | True |

Table 6: Hyperparameter space explored for protein function prediction

| Learning Rate
Hyperparameter Name | $[10^{-1}, 10^{-2}, 10^{-3}, 10^{-4}, 10^{-5}]$
Values Explored |
|---|---|
| Embedding Size | [64, 128, 256] |
| Middle Layer Size | [128, 256, 512] |
| Keep Probability | [0.5, 0.7, 0.8, 0.9, 1.0] |

Table 7: Best hyperparameters for protein function prediction

| Hyperparameter Name | Flat Sigmoid | Bayesian Network of Sigmoids |
|---|---|---|
| Learning Rate | $10^{-3}$ | $10^{-3}$ |
| Embedding Size | 64 | 128 |
| Middle Layer Size | 512 | 512 |
| Keep Probability | 0.7 | 0.7 |

## A.2 BINNED PER-LABEL PERFORMANCE NUMBERS FOR LESS FREQUENT LABELS

Table 8: AUROC results for binned per-label performance on labels with at most 1,000 positive examples for the small disease prediction task.

| Number of Positive Examples | Model | | |
|---|---|---|---|
| | Simple Baseline | Flat Sigmoid | Bayesian Network of Sigmoids |
| 5-10 | 0.66 (0.65-0.66) | 0.68 (0.68-0.68) | **0.77** (0.77-0.77) |
| 5-10 | 0.66 (0.65-0.66) | 0.68 (0.68-0.68) | **0.77** (0.77-0.77) |
| 11-25 | 0.74 (0.74-0.74) | 0.70 (0.70-0.70) | **0.79** (0.79-0.80) |
| 26-50 | 0.78 (0.78-0.79) | 0.72 (0.72-0.72) | **0.81** (0.80-0.81) |
| 51-100 | 0.80 (0.79-0.80) | 0.74 (0.74-0.74) | **0.81** (0.80-0.81) |
| 101-250 | **0.81** (0.81-0.81) | 0.76 (0.76-0.76) | **0.81** (0.81-0.81) |
| 251-500 | **0.82** (0.82-0.82) | 0.77 (0.77-0.77) | 0.81 (0.80-0.81) |
| 501-1000 | **0.83** (0.83-0.83) | 0.80 (0.79-0.80) | 0.82 (0.82-0.82) |

Table 9: Average precision results for binned per-label performance on labels with at most 1,000 positive examples for the small disease prediction task.

| Number of Positive Examples | Model | | |
|---|---|---|---|
| | Simple Baseline | Flat Sigmoid | Bayesian Network of Sigmoids |
| 5-10 | 0.09 (0.08-0.09) | 0.06 (0.06-0.07) | **0.23** (0.22-0.23) |
| 11-25 | 0.11 (0.10-0.11) | 0.04 (0.04-0.04) | **0.19** (0.19-0.19) |
| 26-50 | 0.10 (0.10-0.11) | 0.04 (0.04-0.04) | **0.15** (0.15-0.15) |
| 51-100 | 0.08 (0.08-0.09) | 0.04 (0.04-0.04) | **0.12** (0.11-0.12) |
| 101-250 | 0.08 (0.08-0.08) | 0.05 (0.04-0.05) | **0.10** (0.09-0.10) |
| 251-500 | 0.08 (0.08-0.08) | 0.06 (0.06-0.06) | **0.09** (0.09-0.09) |
| 501-1000 | 0.10 (0.10-0.10) | 0.09 (0.09-0.09) | **0.11** (0.11-0.11) |

Table 10: AUROC results for binned per-label performance on labels with at most 1,000 positive examples for the large disease prediction task.

| Number of Positive Examples | Model | | |
|---|---|---|---|
| | Simple Baseline | Flat Sigmoid | Bayesian Network of Sigmoids |
| 5-10 | 0.45 (0.42-0.48) | 0.69 (0.66-0.72) | **0.71** (0.68-0.73) |
| 11-25 | 0.50 (0.48-0.51) | 0.75 (0.74-0.77) | **0.76** (0.74-0.78) |
| 26-50 | 0.67 (0.65-0.68) | **0.79** (0.77-0.80) | **0.79** (0.77-0.80) |
| 51-100 | 0.76 (0.76-0.77) | 0.80 (0.80-0.81) | **0.81** (0.80-0.82) |
| 101-250 | 0.81 (0.81-0.82) | **0.82** (0.82-0.83) | **0.82** (0.82-0.82) |
| 251-500 | 0.83 (0.82-0.83) | 0.83 (0.83-0.84) | **0.84** (0.83-0.84) |
| 501-1000 | **0.85** (0.85-0.85) | **0.85** (0.85-0.85) | **0.85** (0.85-0.85) |

Table 11: Average precision results for binned per-label performance on labels with at most 1,000 positive examples for the large disease prediction task.

| Number of Positive Examples | Model | | |
|---|---|---|---|
| | Simple Baseline | Flat Sigmoid | Bayesian Network of Sigmoids |
| 5-10 | 0.08 (0.05-0.11) | 0.33 (0.28-0.37) | **0.38** (0.34-0.43) |
| 11-25 | 0.11 (0.10-0.13) | 0.38 (0.36-0.41) | **0.43** (0.40-0.46) |
| 26-50 | 0.25 (0.23-0.27) | 0.42 (0.39-0.44) | **0.45** (0.43-0.47) |
| 51-100 | 0.33 (0.32-0.35) | 0.39 (0.37-0.40) | **0.43** (0.42-0.45) |
| 101-250 | 0.35 (0.34-0.36) | 0.36 (0.35-0.37) | **0.39** (0.38-0.39) |
| 251-500 | 0.31 (0.30-0.32) | 0.32 (0.31-0.33) | **0.35** (0.35-0.36) |
| 501-1000 | 0.29 (0.29-0.30) | 0.29 (0.28-0.29) | **0.32** (0.31-0.32) |

Table 12: AUROC results for binned per-label performance on labels with at most 1,000 positive examples for the protein function prediction task.

| Number of Positive Examples | Model | | |
|---|---|---|---|
| | Simple Baseline | Flat Sigmoid | Bayesian Network of Sigmoids |
| 5-10 | 0.62 (0.60-0.63) | 0.73 (0.71-0.75) | **0.80** (0.79-0.82) |
| 11-25 | 0.61 (0.60-0.62) | 0.74 (0.72-0.76) | **0.80** (0.79-0.82) |
| 26-50 | 0.63 (0.61-0.64) | 0.77 (0.75-0.79) | **0.80** (0.78-0.82) |
| 51-100 | 0.63 (0.62-0.65) | 0.78 (0.76-0.79) | **0.81** (0.79-0.82) |
| 101-250 | 0.64 (0.63-0.65) | 0.77 (0.75-0.78) | **0.79** (0.78-0.81) |
| 251-500 | 0.64 (0.63-0.65) | 0.76 (0.75-0.78) | **0.78** (0.77-0.79) |
| 501-1000 | 0.65 (0.64-0.66) | 0.75 (0.74-0.77) | **0.77** (0.76-0.78) |

Table 13: Average precision results for binned per-label performance on labels with at most 1,000 positive examples for the protein function prediction task.

| Number of Positive Examples | Model | | |
|---|---|---|---|
| | Simple Baseline | Flat Sigmoid | Bayesian Network of Sigmoids |
| 5-10 | 0.19 (0.17-0.22) | 0.16 (0.14-0.18) | **0.23** (0.21-0.26) |
| 11-25 | 0.15 (0.13-0.17) | 0.18 (0.17-0.20) | **0.21** (0.19-0.23) |
| 26-50 | 0.13 (0.11-0.15) | **0.21** (0.18-0.23) | **0.21** (0.19-0.23) |
| 51-100 | 0.12 (0.11-0.14) | **0.21** (0.19-0.23) | 0.20 (0.18-0.22) |
| 101-250 | 0.12 (0.10-0.13) | **0.21** (0.19-0.23) | 0.21 (0.20-0.23) |
| 251-500 | 0.12 (0.11-0.14) | **0.22** (0.20-0.24) | **0.22** (0.20-0.23) |
| 501-1000 | 0.16 (0.15-0.18) | **0.27** (0.25-0.29) | **0.27** (0.25-0.29) |

## A.3 Protein Reweighted Flat Sigmoid Baseline Binned Per-Label Performance Numbers For Less Frequent Labels

Table 14: AUROC results for binned per-label performance on labels with at most 1,000 positive examples for the protein function prediction task with the reweighted flat sigmoid baseline.

| Number of Positive Examples | Reweighted Flat Sigmoid |
| --- | --- |
| 5-10 | 0.76 (0.74-0.77) |
| 11-25 | 0.76 (0.74-0.78) |
| 26-50 | 0.77 (0.76-0.79) |
| 51-100 | 0.79 (0.77-0.80) |
| 101-250 | 0.78 (0.76-0.79) |
| 251-500 | 0.77 (0.76-0.79) |
| 501-1000 | 0.77 (0.75-0.78) |

Table 15: Average precision results for binned per-label performance on labels with at most 1,000 positive examples for the protein function prediction task with the reweighted flat sigmoid baseline.

| Number of Positive Examples | Reweighted Flat Sigmoid |
| --- | --- |
| 5-10 | 0.14 (0.12-0.16) |
| 11-25 | 0.16 (0.14-0.18) |
| 26-50 | 0.17 (0.17-0.22) |
| 51-100 | 0.22 (0.19-0.23) |
| 101-250 | 0.22 (0.20-0.24) |
| 251-500 | 0.22 (0.21-0.24 |
| 501-1000 | 0.27 (0.25-0.29) |

