# OpenReview forum: "Using Ontologies To Improve Performance In Massively Multi-label Prediction"
_ICLR.cc/2019/Conference_

### Official Review · AnonReviewer1 · 2018-11-01
**Using label structure to address class imbalance**

**Rating:** 4
**Confidence:** 4

**Review:**

This is a clear and well written paper that attempts to improve our ability to predict in the setting of massive multi-label data which, as the authors highlight, is an increasingly import problem in biology and healthcare.

Strengths:
The idea of using the hierarchical structure of the labels is innovative and well-motivated. The experimental design and description of the methods is excellent.

Weaknesses:
Overall the results are not consistently strong and there is a key baseline missing. The approach only seems help in the "rare label, small data" regime, which limits the applicability of the method but is still worthy of consideration.

My biggest reservation is that the authors did not include a baseline where the classes are reweighted according to their frequency. Multilabel binary cross-entropy is very easy to modify to incorporate class weights (e.g. upweight the minority class for each label) and without this baseline I am unable to discern how well the method works relative to this simple baseline.

One more dataset would also strengthen the results, and since I am suggesting more work I will also try to be helpful and be specific. Predicting mesh terms from abstracts would qualify as a massive multilabel task and there is plenty of public data available here: https://www.nlm.nih.gov/databases/download/pubmed_medline.html

Finally, there is one relevant paper that the authors may wish to consider in their review section: https://www.biorxiv.org/content/early/2018/07/10/365965

---

> ### Author Response · Authors · 2018-11-22
> **Added additional baseline you suggested**
>
> Hi,
>
> Thank you for the review and for suggesting an inverse frequency-weighted baseline. That’s a very nice/simple idea that has the potential for improving the performance on rare labels. We implemented the standard 1/sqrt weighting scheme (used in other works such as Word2Vec) and re-performed our hyperparameter search for the protein models. We have noted this baseline within the paper and included the exact performance table in the appendix.
>
> The reweighting does seem to somewhat improve performance on some of the codes (particularly those that are of medium frequency or above). However, it does not seem to help much in the more rare scenarios that we are investigating. The main issue is that these codes are rare enough that we believe that there are too few examples to even train the final classification layer.  It’s possible that increased weight on those labels simply encourages more overfitting on those few examples.
>
> The MeSH dataset certainly sounds interesting and seems to be a nice public multi-label dataset. Unfortunately, we don’t have the resources to run our experiments on that dataset for this paper. We already have two different datasets with a total of three distinct setups. However, we will definitely keep that dataset in mind for future multi-label work!

---

### Official Review · AnonReviewer3 · 2018-11-04
**Lack experiment comparison with previous work. The experiment results don't include any significance yet.**

**Rating:** 5
**Confidence:** 3

**Review:**

The authors propose a new training scheme for training neural networks for multi-label prediction tasks by introducing ontology relationships between labels.
The paper motivates very well by the observations that some labels include very small amount of data points.
However, the authors don’t really investigate why such labels are rarely observed and the experiments don’t include any significance.
Thus overall I don’t think the paper is ready for publishing for ICLR yet.

Below are some more detailed comments:
1) The authors discuss nicely about the intuition to introduce the Bayesian networks in the tasks of disease prediction. Essentially, the probability of assigning the label (leaf node) should be account for the probability of it being observed, namely the prior. Thus it is not surprising that for the rare labels, the proposed method would yield higher precision. However, the experiments don’t really include any significance measurement; especially for such tasks where the number of testing examples with rare labels is small (5~10 positive examples), significance measurement or some forms of hypothesis testing is a must-have in order to draw conclusion about the performance comparison. Answering such significance issue with tests for overfitting would be nice.

2) My other major concern is for the protein function prediction task, the reason of why for certain labels, the number of instances is small, could be due to that a) there don’t exist much biological web-lab evidence, or b) among the population, there indeed only exist small number of proteins associated with such labels. The proposed method can address b) but not necessarily address a).

3) The paper discusses the other results very briefly in Section 5 but doesn’t include any experiment comparison. Thus it is not convincing that the proposed method is making contribution to the field of disease prediction, protein function prediction or even general multi-label prediction. I would suggest to include the comparison with the state of the art methods for each application.

4) Particularly for protein function prediction, another line of studies is to use protein protein interaction networks or other sources such as functional pathways rather than using sequence information alone (ref below). Some discussion would be nice.

Schwikowski, Benno, Peter Uetz, and Stanley Fields. "A network of protein–protein interactions in yeast." Nature biotechnology 18.12 (2000): 1257.
Cao, Mengfei, et al. "New directions for diffusion-based network prediction of protein function: incorporating pathways with confidence." Bioinformatics 30.12 (2014): i219-i227.

---

> ### Author Response · Authors · 2018-11-22
> **Added confidence intervals, other clarifications**
>
> Hi,
>
> Thank you for the review.
>
> 1) Per your recommendation, we have computed bootstrap 95% confidence intervals that measure the variability of our results with respect to the test set. We have included said confidence intervals in the tables within the appendix. In general, even though there are a very small number of positive examples per label, the vast number of labels enables us to compute performance differences between methods with a certain amount of precision.
>
> 2) You are correct that we do not currently deal with the issue of missing labels. Missing labels are definitely present in these datasets, as many protein annotations simply aren’t recorded and many diseases aren’t diagnosed (or at the very least coded). Our work is primarily focused only on improving the predictive power for the rare labels given the data we have. Exploring the missing label problem would be an interesting future extension and we do believe that this work might be able to help with those efforts.
>
> 3) As for the baselines, our focus is to compare the bayesian network modification compared to the flat-sigmoid approach, which is the standard approach used in state-of-the-art methods. It is difficult to exactly replicate others’ datasets, but the baseline was tuned equally heavily to provide a fair comparison.
>
> 4) We are aware that there are features such as protein interaction networks that can improve prediction performance for specific tasks, but the focus of the paper is not protein function prediction. Rather, the focus is on a generic machine learning algorithm that could be applied to many non-biological tasks.

---

### Official Review · AnonReviewer2 · 2018-11-05
**Multi-label using ontology**

**Rating:** 6
**Confidence:** 3

**Review:**

This paper proposes a neural network, the outputs of which create a "Bayesian network of sigmoids". This is for use in massively multi-label situations where the class outputs are connected to some ontology. By using the ontology, the performance on long-tail classes with few examples should be improved.
I like the method. It is intuitive and easy to implement. The only issue I have with the features and model is when in 3.1.2 the weights for the medical labels as input to the encoder are said to be 'tied' to the output label embeddings. I would like to have seem more justification for this (perhaps just the keeping number of parameters down?) or evaluation as to whether it helped - I think it may have hindered because of how the output label is used in the dot product.
The stated difference between this work and previous hierarchical softmax models is that they use DAG structures, not just tree structures. However, of the two datasets they try, only one (proteins) has a DAG structure to the ontology, and they do not have a baseline comparison with a hierarchical softmax model.
The method does show improvements at low number of examples against a flat sigmoid model in the small disease prediction and protein function prediction, but other results are mixed, especially for the proteins with the DAG ontology, which is where one would have liked to see an advantage.
I would like to have seen performance against a hierarchical soft max framework, or on some openly available or benchmark datasets, otherwise it is hard to judge the utility of the method.

---

> ### Author Response · Authors · 2018-11-22
> **Explaining tied embeddings + relationship to hierarchical softmax**
>
> Hi,
>
> Thank you for reviewing our paper. We tied the input embeddings to the output embeddings because it improved the results for both our baseline and Bayesian models. We didn’t bother running the full hyperparameter grid for both tied and untied embeddings because it seemed like a clear win and it wasn’t the main point of the paper.
>
> We did not include a hierarchical softmax baseline in this work because hierarchical softmax applies only to multi-class problems while we focus on multi-label problems. In fact, this work can be seen as an extension of hierarchical softmax to handle multi-label settings (with the additional support of DAGs added in as DAGs make much more sense in a multi-label setting as opposed to a multi-class setting).
>
> As for replicability, the protein dataset we are using is fully public. After review, we plan on publicly posting our cleaned splits and protocol buffers to encourage comparisons.

---

### Author Response · Authors · 2018-11-22
**Summary of changes**

We thank the reviewers for the feedback which we have incorporated into the revised version of the paper. The biggest changes are:

  1. Added confidence intervals (derived from bootstrapping the test set) to result tables to better show the significance of differences.
  2. Added a baseline that consisted of a flat sigmoid model combined with per-code weights to upweight less frequent codes.

We also clarified points of confusion in reviewer-specific responses below.

---

### Meta-Review · Area_Chair1 · 2018-12-19
**Nice approach, more convincing empirical evaluation may be needed**

**Confidence:** 4
**Recommendation:** Reject

**Metareview:**

The paper proposes a nice approach to massively multi-label problems with rare labels which may only have a limited number of positive examples; the approach uses Bayes nets to exploit the relationships among the labels in the output layer of a neural nets. The paper is clearly written and the approach seems promising, however, the reviewers would like to see even more convincing empirical results.